# Implementation and effectiveness outcomes of Community Health Advocacy Teams to improve long-lasting insecticide net distribution and use in six districts in Ghana: A one-group pre-post-test study

**Franklin N. Glozah**[1], **Philip Teg-Nefaah Tabong**[1] *, **Eva Bazant**[2], **Emmanuel Asampong**[1], **Ruby Hornuvo**[1], **Adanna Nwameme**[1], **Nana Yaw Peprah**[3], **Gloria M. Chandi**[4], **Philip Baba Adongo**[1], **Phyllis Dako-Gyeke**[1]

1 Department of Social and Behavioural Sciences, School of Public Health, University of Ghana, Accra, Ghana, 2 Health Campaign Effectiveness Coalition, Task Force for Global Health, Decatur, Georgia, United States of America, 3 National Malaria Elimination Programme, Accra, Ghana, 4 Ghana Health Service, Ga North Municipal Health Directorate, Greater Accra, Ghana

* ptabong@ug.edu.gh

## Abstract

Malaria remains a leading cause of illness and death especially among children and pregnant women in Ghana. Despite the efforts made by the National Malaria Elimination Programme (NMEP), including distribution of Long-Lasting Insecticide Nets (LLINs) to households through periodic Point Mass Distribution (PMD) campaigns and continuous channels (antenatal, schools and postnatal), there is a gap between access and use of LLINs in Ghana. An effective and functional community-based group that would seek to improve the effectiveness of LLIN distribution before, during, after PMD Campaigns and continuous distribution at the community level could help address this gap. This paper assesses the implementation outcomes and short-term effectiveness of the pilot implementation of co-created community health advocacy teams (CHAT) intervention in Ghanaian communities to plan and implement campaigns to increase LLIN use. The study employed a one-group pre-post study design and measured implementation outcomes (acceptability, appropriateness, and feasibility) and effectiveness outcomes (LLIN awareness, LLIN access, willingness to purchase LLIN, and LLIN use) among 800 community households. The CHAT intervention was implemented for four months across six districts in the Eastern and Volta regions of Ghana. The data were downloaded directly from REDCap and analyzed statistically (descriptive and McNemar test of association) using SPSS 22 software. After the implementation period, the majority of respondents in all six districts indicated that the CHAT intervention was acceptable (89.8%), appropriate (89.5%), and feasible (90%). Also, there was a significant association between baseline and end-line assessment on all four effectiveness outcome measures. Household members' awareness of, access to, willingness to purchase, and use of LLINs increased significantly over the four-month period that the CHAT intervention was implemented. The study concludes that CHAT is an

**Data Availability Statement:** All data and related metadata underlying the findings reported can be obtained from the Ghana Health Service Ethics Review Committee on ethics.research@ghsmail.org.

**Funding:** This work was supported in whole by the Bill & Melinda Gates Foundation (Grant Number INV-01076 to the Task Force for Global Health's Health Campaign Effectiveness (HCE) Program) and received by PDG, FNG, PT-NT, EA, AN, GMC, and PBA. The funders had no role in study design, data collection and analysis, decision to publish, or preparation of the manuscript.

**Competing interests:** The authors have declared that no competing interests exist.

acceptable, appropriate, and feasible intervention for supporting the National Malaria Programme in LLIN PMD and for engaging in Social and Behaviour Change Communication activities through the continuous channels of distribution. Additionally, the CHAT demonstrates short-term effectiveness outcomes in terms of creating LLIN awareness, providing access to LLIN, and encouraging Ghanaian community members to be willing to purchase and use LLINs. Although the activities of CHAT members were largely voluntary, integration into the existing primary health care system will make it sustainable.

## Background

Globally, there were an estimated 247 million cases of malaria and 619,000 malaria deaths in 2021, with the WHO African Region recording a relatively high malaria burden. 95% of malaria cases and 96% of malaria deaths among children under 5 years of age, were recorded accounting for 80% of malaria mortality in the in the WHO African Region [1]. Ghana is one of the 15 countries in Africa with the highest malaria burden, with 2.1% of malaria cases and 1.9% of malaria deaths. From 2017 to 2020, about 4.3% of West African malaria cases in were in Ghana [2]. Malaria remains a major cause of illness and death in Ghana, especially among children under five years of age and pregnant women, and accounts for 41% of suspected outpatient, 21% confirmed, and 18% of inpatient malaria cases in 2020.The disease continues to be the most expensive for the National Health Insurance Scheme (NHIS) to cover and has a huge impact on all aspects of human life [3].

The National Malaria Elimination Programme (NMEP) is responsible for reducing malaria morbidity and mortality in Ghana. Over the years, the programme has carried out several malaria prevention interventions such as the Point Mass Distribution (PMD) of Long-Lasting Insecticide-Treated Nets (LLINs) [3]. The distribution and use of LLINs are core interventions for preventing malaria infection in malaria-endemic countries, including Ghana. LLINs provide protection against mosquito bites, repel, and kill mosquitoes, thereby reducing the transmission of malaria parasites and decreasing malaria risk at the individual and community levels when high coverage is achieved [4]. The 2014–2020 Ghana Strategic Plan for Malaria Control focused on scaling up LLIN use and other malaria interventions including ownership and use of LLINs, indoor residual spraying (IRS), and intermittent preventive treatment during pregnancy using sulfadoxine and pyrimethamine (IPTp-SP)] to reduce the malaria morbidity and mortality burden by 75% by 2020 [4].

LLINs can be obtained mainly during PMD campaigns; however, as part of targeted continuous distribution programmes, LLINs are distributed through antenatal care (ANC), child welfare clinics (CWC), and primary schools. During their initial ANC visit, pregnant women are the focus of distribution in health facilities. The LLINs are given out free to children 18 months or older as part of child welfare visits, at the time of their second dose of the measles-rubella vaccine [4]. Children in Primary 2 and Primary 6 in public and private schools across the country receive free LLINs as part of school-based distributions in years when a PMD campaign does not take place [4].

Even though over the years there have been improvements in overall LLIN ownership, NMEP's strategic goal of 80% utilisation among pregnant women and children under five is yet to be met. The 2019 Ghana Malaria Indicator Survey shows that 67% of Ghanaian households have LLINs (access), but only 43% of the Ghanaian household population slept under LLINs the night before the survey. This indicates that a relatively large number of people have

not slept under the LLIN despite the distribution campaign. These distribution efforts have exposed a high proportion of Ghanaians to having at least one LLINs at the household; however, this has not translated to commensurate LLIN use. Among other reasons, heat from sleeping in the LLIN is one of the most cited barriers to regular LLIN use, especially during the dry season. Other reported barriers to LLIN use include skin irritation; poor ventilation in sleeping areas; and, in some cases, a lack of knowledge of the relationship between the LLIN use and malaria prevention [5,6].

The LLINs distributed through campaigns and continuous distribution channels are normally provided for free using the formula two household members to one net. However, with declining support to the country as it attained a middle-income status, there is the need to position users to begin to think about procuring their own LLINs. LLINs are socially desired and accepted in many communities, but programmes struggle to convince people to buy, maintain, and utilise them properly. Designing a sustainable and effective LLINs strategy is difficult because several social and cultural factors influence the adoption and use of LLINs in communities. For example, in several parts of Africa, including Ghana, malaria health-seeking behaviour among people does not indicate a connection between mosquito bites and malaria as these are attributed to other factors [7,8].

In addition to the above, some barriers to LLIN use including limited social and behaviour change communication (SBCC) activities, lack of continuous malaria education, knowledge gap on malaria prevention, inability to hang LLINs in many households due to housing type and sleeping places, as well as misuse and repurposing of LLINs have been identified and documented in numerous studies [9]. These suggest that more social, cultural, and behavioural research is needed to understand how local knowledge of transmission, diagnosis, treatment, and prevention influences the utilisation of LLINs interventions. In order to create community-based malaria control programmes that are sustainable and encourage behaviour change as well as the adoption of new concepts and technologies, it is helpful to understand local knowledge about malaria [7] and used of context specific strategies.

## The Community Health Advocacy Team (CHAT) innovation in Ghana

Evaluations conducted in Ghana have consistently discovered a large disparity between household ownership of LLINs and the use of LLINs, with LLINs use always significantly lower than ownership of at least one LLINs. This has been interpreted as evidence of the inability to achieve appropriate LLINs use, or as a failure of social behaviour change communication (SBCC) to sufficiently improve LLIN use [6–8,10,11]. Considering this evidence, innovative social interventions that encourage behaviour change are required to meet the objectives of the LLIN distribution campaign [12]. Through community-based programmes, government, health organisations, and social actors, all relevant stakeholders can collaborate closely to solve public health problems in the population.

As a consequence, a Community Health Advocacy Team (CHAT) intervention was co-created through the participatory learning in action technique using participatory workshops (PWs) involving stakeholders (researchers, community members, district health management team and the NMEP) in six study districts in Ghana. This was an initiative by the study team from the University of Ghana School of Public Health in collaboration with the NMEP.

To co-create the CHAT intervention, findings from the initial phases of the project (i.e., desk review, focus group discussions (FGDs), key informant interviews (KIIs), and baseline surveys) were synthesized and grouped according to relevance and distilled which formed the basis for developing a PW guide. This guide was then used for the moderation of the PWs

aimed at cocreating a LLIN campaign intervention involving various stakeholders. Findings from the PWs suggested the establishment of a CHAT can be instrumental in facilitating and improving the effectiveness of LLIN distribution campaigns within communities in Ghana [9]. The CHAT uses a person-centred approach to promote LLIN access and use by leveraging on Ghana's Community-based Health Planning and Services (CHPS) programme (i.e., primary health care system), to ensure community involvement, ownership, and sustainability of the LLIN PMD campaigns [9].

A CHAT consists of nine members who are influential in their respective communities: namely, Community Health Officers, religious leaders, School Health Education Programme coordinators, assemblymen/women, community information officers, representatives from the security services, community-based organisations, and traditional authorities. The terms of reference of the CHAT are generally based on NMEP's key elements of the campaign functions at the sub-district level: household registration, training, SBCC activity, logistics, distribution, and supervision. The CHAT meetings are convened quarterly, preferably by a CHO.

The CHAT members were trained on how to carry out malaria education and prevention activities, as well as the promotion of LLINs use within communities and primary health care levels during and after PMD campaigns. Specifically, the CHAT members are expected to provide community-level support for LLIN distribution channels through PMD campaigns; through continuous distribution of LLINs in schools, ANC and CWC; and develop context-based Social and Behaviour Change Communication (SBCC) strategies on malaria prevention and consistent use of LLINs. The CHAT sensitizes the community on the proper use and maintenance of LLINs, supports with the management of LLINs logistics during distribution exercises and accountability, and assists other community-based health campaigns [13]. CHAT use community-based strategies such as house-to-house, community durbars, use community information centres and school health platforms to promote the use of LLINs. In addition, CHAT members collaborate with district health management team and other civil society groups to provide voluntary campaigns on LLINs use. The teams officially started their activities in the community in November 2021.

This paper assesses the implementation outcomes and short-term effectiveness of the co-created CHAT intervention in six districts in Ghana. We hypothesise that if the CHATs are effective after a four-month pilot implementation, the CHAT will be highly acceptable, appropriate, and feasible, and there will be a significant increase in LLIN awareness, access, willingness to purchase, and use before and after the pilot implementation. Although both qualitative and quantitative methods were used at various stages of the study to contextualize and explore the barriers to, and enablers of Mass LLIN Distribution Campaigns, and to identify baseline parameters to be used for assessing the effectiveness of our co-created intervention, only the outcomes of the quantitative surveys are presented in this paper.

## Methods and materials

### Ethical approval

Ethical clearance was obtained from the Ghana Health Service Ethics Review Committee before the commencement of all data collection. All research assistants received specific training before data collection per the study's training protocol. Participants involved in this study provided written informed consent before participating in the study. Participants were informed about the aim of the study, procedures, benefits of the study, as well as their rights as participants. The information and consent documents for participants were written in simple English. To enhance comprehension, research assistants were present during the informed consent process to explain any questions that the participants did not understand. Those

consenting to participate in the study signed (or placed a thumbprint on) an informed consent form, before participating in the study. All participants were assured that the information they provided would be handled confidentially and research findings would be reported with complete anonymity.

## Study design

We used a one-group pre-post study design and measured our outcomes (1) LLIN awareness [2], LLIN access [3], willingness to purchase LLIN [4], LLIN use [5], acceptability [6], appropriateness, and [7] feasibility among community members, before implementing the CHAT intervention, and again four months after implementation. We used the Reach, Effectiveness, Adoption, Implementation and Maintenance (RE-AIM) framework [14] as the basis for determining and assessing the implementation outcomes and short-term effectiveness of our study outcomes. Using the RE-AIM framework, we examined the Reach of the communication in terms of who are expected to benefit from the Campaigns and the percent of those are actually exposed to intervention. The Effectiveness of the Campaign was measured to determine the contribution of the Campaign to regular use of LLIN among the targeted population for the Campaign. We also measured the context (Adoption) in which the communication Campaigns apply and determined how the communication Campaigns were applied and the level of compliance with what was conceptualized (Implementation). Finally, we explored the sustainability of the Campaign strategies. Tools developed by the National Working Group on RE-AIM Planning and Evaluation Framework [14] and the acceptability framework created by Sekhon and colleagues [15] were adapted to collect quantitative data.

To assess effectiveness, we restricted ourselves to measuring short-term differential rates in the utilisation of the CHAT and its related activities.

## Study areas

The study was conducted in six districts–three in the Eastern Region and three in the Volta Region of Ghana (Fig 1). These were communities in districts where the 2021 PMD campaigns of LLINs were ongoing. We ensured that baseline assessments were done in the selected communities before their involvement in NMEP's key elements of the campaign (i.e., registration and distribution activities for the 2021 PMD). Also, the implementation of the CHAT was planned to run within NMEP structures as our aim was to transition the entire PMD campaign into CHPS. The districts that participated in the study were selected through purposive sampling because they recorded the highest malaria test positivity rate among patients attending health facilities (as indicated in their percentages) in 2020 in the two regions: Ho West (Tsito-90%), Ho (Takla Hokpeta-75%) and Agortime Ziope (Kpetoe-100%) in the Volta Region; and Birim South (Apoli-94%), Achiase (Achiase-94%) and Abuakwa North (Kukurantumi-93%) in the Eastern Region [16]. At the time of the study, continuous/routine LLINs distributions in schools and antenatal clinics were also ongoing in these communities.

## Population, sample size and sampling procedure

The study population was made up of adult men and women from communities within the six districts in the two regions in southern Ghana. We used data from the Ghana Statistical Service (GSS) to initially list all eligible households in the selected communities. A systematic random sampling technique was then used to select the needed number of households for the study. Each community in the targeted region and district was allocated a sample size proportional to its population size. Using Cochran's Sample Size Formula with an $\alpha$-level of 5%, a margin of error associated with the estimates to be 5%, and a 50% anticipated increase in use of Mass

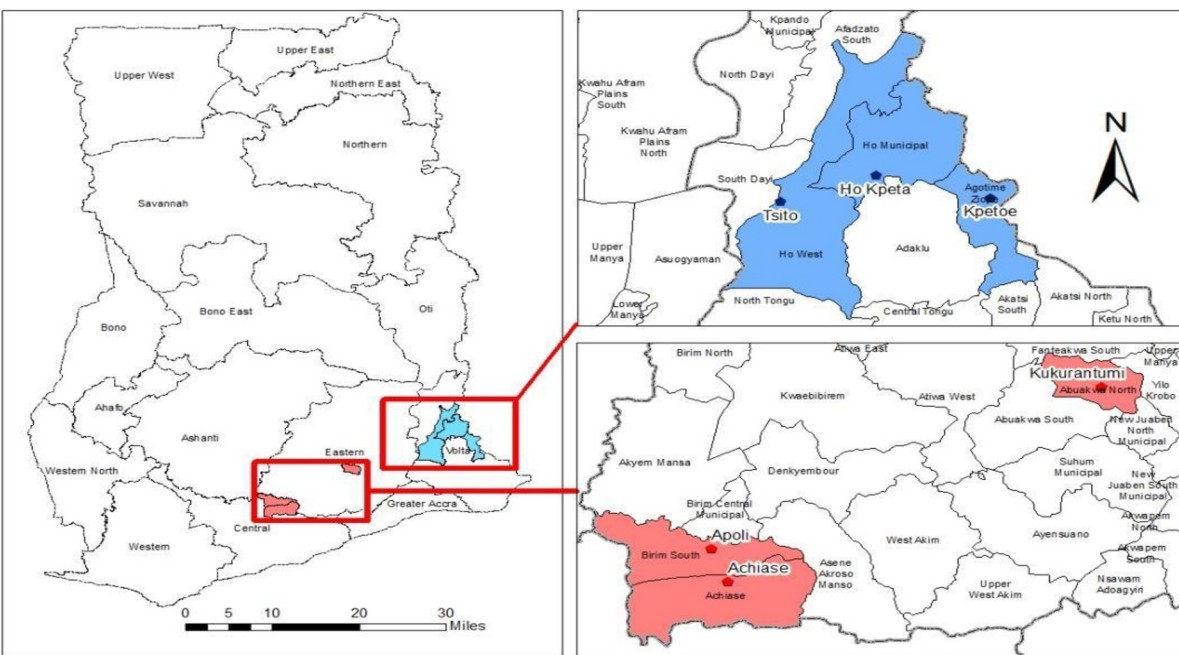

**Fig 1. Location of the six districts in the Eastern and Volta regions of Ghana.** Source: Created with ArcGIS (basemap shapefile: Ghana Statistical Service: https://data.humdata.org/dataset/cod-ab-gha).

LLIN distribution between baseline and endline, the estimated sample size was 800 for both baseline and end-line surveys.

## Data collection tools and procedure

A structured questionnaire was developed for both the baseline and end-line surveys to collect information on the socio-demographic characteristics of households, socioeconomic status of the households, knowledge and perceptions about co-created intervention, use of LLINs, care of LLINs, the prevalence of malaria among children under five six weeks before the survey, etc. The survey questionnaire was developed in Research and Electronic Data Capture (REDCap) for onsite electronic data collection. The structured questionnaires were translated into three key local languages (Twi, Ga, and Ewe) and preloaded on Android tablet computers. All data were collected using face-to-face interviews with trained research assistants. Responses were uploaded instantly or shortly afterwards into a cloud-based data management system, which was regulated by the project team at the University of Ghana School of Public Health. In the baseline survey, participants were recruited from 21st August to 5th September 2021. During the endline, participants were recruited from 24th March-7th April 2022.

## Measures

**Implementation outcomes.** *Measures of a*cceptability, appropriateness, and feasibility were used to assess implementation outcomes of our intervention implementation. Acceptability is the perception among implementation stakeholders that a given treatment, service, practice, or innovation is agreeable, palatable, or satisfactory; appropriateness is the perceived fit, relevance, or compatibility of the innovation or evidence-based practice for a given practice setting, provider, or consumer, and/or perceived fit of the innovation to address a particular issue or problem; while feasibility is defined as the extent to which a new treatment or

innovation can be successfully used or carried out within a given agency or setting [17]. These measures have four items each on a five-point Likert scale ranging from never (0) to always (4). In a validation study, the Cronbach's alpha for the acceptability, appropriateness, and feasibility measures were 0.85, 0.91 and 0.89 respectively [17].

**Short-term effectiveness outcomes.** Awareness, access to LLIN, willingness to purchase LLIN, and LLIN use were utilised to assess the short-term effectiveness of the implementation of our intervention. These measures were adapted from the 2019 Ghana Malaria Indicator Survey Household Survey [4]. Awareness was assessed with the question "Have you heard about mosquito/bed net?"; access was assessed by asking "Does your household have a mosquito/bed net?"; willingness to purchase was assessed by asking "Are you willing to buy mosquito/bed nets for your household members?"; and LLIN use was assessed with the question "Did you or any of your household members sleep under a mosquito/bed net last night?" All four questions had dichotomized response options, Yes or No.

In addition to the implementation outcome measures and short-term effectiveness measures, time (i.e., baseline and end-line) was used as the main predictor of the effectiveness outcomes.

**Statistical analysis.** Data were downloaded directly from REDCap into SPSS 22. Data editing and formatting were done to ensure the correctness of the data. Data were also explored for normality using skewness and kurtosis prior to analysis. To estimate the level of acceptability, appropriateness, and feasibility of the intervention, we first created summated scores of these measures and then categorised them into low and high. Only completed data out of the 800 were used to compute the summated scores and for further analysis. Descriptive statistics were then performed to estimate the frequencies of the implementation outcome variables at end-line only. Also, the McNemar test for paired proportions was performed to determine the association between baseline and end-line measures of all short-term effectiveness outcomes i.e., LLIN awareness, access, willingness to purchase and use. A p-value $<0.05$ was considered statistically significant.

## Results

### Socio-demographic characteristics of participants

At both baseline and end-line, females were 70% of participants with a near equal number of males and females in all six districts. There were some few differences between the baseline and endline figures observed (i.e., Birim South and Achiase) and this was due to the unavailability of some participants (n = 2) who participated in the baseline hence any other available member of the same household who met the selection criteria were involved. (Table 1)

### Assessment of implementation outcomes

Nearly all (90%) of community members thought that the intervention was acceptable, appropriate, and feasible after its implementation (Table 2).

### Assessment of short-term effectiveness outcomes

There was a statistically significant association between baseline and end-line assessment on all four effectiveness outcome measures (Table 4). Household members' awareness of LLINs, access to LLINs, willingness to purchase LLINs, and use of LLINs increased significantly over the four-month period that the CHAT intervention was implemented in the six districts. Households in the six districts were more likely to engage with LLIN campaign activities after the implementation of the CHAT intervention.

**Table 1. Socio-demographic characteristics of participants.**

| District | Sex | | | | | |
|---|---|---|---|---|---|---|
| | Baseline (N = 800) | | | Endline (N = 800) | | |
| | Male | Female | Total | Male | Female | Total |
| Ho West (Tsito)[1] | 40 | 95 | 135 | 40 | 95 | 135 |
| Agotime Ziope (Kpetoe)[1] | 28 | 107 | 135 | 28 | 107 | 135 |
| Ho (Hokpeta)[1] | 43 | 87 | 130 | 43 | 87 | 130 |
| Birim South (Apoli)[2] | 58 | 70 | 128 | 57 | 71 | 128 |
| Achiase (Achiase)[2] | 36 | 102 | 138 | 37 | 101 | 138 |
| Abuakwa North (Kukurantumi)[2] | 36 | 98 | 134 | 36 | 98 | 134 |
| **Total** | **241** | **559** | **800** | **241** | **559** | **800** |

[1]Volta Region

[2]Eastern Region.

The highest increase was in access, followed by use, willingness to purchase, and then awareness.

## Discussion

This paper assesses the implementation outcomes and short-term effectiveness of the co-created CHAT intervention for awareness, access, willingness to purchase, and use of LLINs across six districts in Ghana. The current study contributes to our understanding of how CHATs can be used to promote LLINs before, during and after campaigns in Ghana.

The study findings show that the CHAT intervention was highly accepted by the community members in all six study districts. This implies that the community members view the CHAT intervention as satisfactory in promoting access to and use of LLIN. Building trust among various stakeholders contributes to the acceptability of implementation research [18]. The engagement of stakeholders and community members throughout all phases of this study have contributed to trust building. Hence, the high acceptability observed here can be attributed to the approach used in this study, such as the use of SBCC, community mobilisation, and engagement of community stakeholders and community members throughout the intervention development and implementation processes of this study. As evident in other studies employing Social and Behaviour Change Communication (SBCC) strategies, the involvement of community stakeholders, leaders and community volunteers helps build trust among community members and fosters community buy-in of malaria intervention which is crucial to the acceptability and sustainability of interventions [19].

Similarly, the study showed that the CHAT intervention was appropriate and feasible. While appropriateness as an implementation outcome refers to the perceived fit, relevance, or

**Table 2. Acceptability, appropriateness, and feasibility of the Community Health Advocacy Team in six districts.**

| Outcome measures | Level | | |
|---|---|---|---|
| | Lower (N (%)) | Higher (N (%)) | Total (N (%)) |
| Acceptability | 36 (10.2) | 317 (89.8) | 353 (100) |
| Appropriateness | 37 (10.5) | 316 (89.5) | 353 (100) |
| Feasibility | 35 (10) | 316 (90) | 351 (100) |

Also, when disaggregated by district, the majority of community members in all six districts mentioned that the CHAT intervention was acceptable, appropriate and feasible (Table 3).

**Table 3. Acceptability, appropriateness, and feasibility of the Community Health Advocacy Team by district.**

| District | Acceptability | | | Appropriateness | | | Feasibility | | |
|---|---|---|---|---|---|---|---|---|---|
| | Lower | Higher | Total | Lower | Higher | Total | Lower | Higher | Total |
| Ho West (Tsito) | 15 | 16 | 31 | 15 | 16 | 31 | 15 | 16 | 31 |
| Agotime Ziope (Kpetoe) | 14 | 33 | 47 | 15 | 32 | 47 | 14 | 33 | 47 |
| Ho (Hokpeta) | 6 | 51 | 57 | 6 | 51 | 57 | 5 | 51 | 56 |
| Birim South (Apoli) | 1 | 101 | 102 | 1 | 101 | 102 | 1 | 102 | 103 |
| Achiase (Achiase) | 0 | 87 | 87 | 0 | 87 | 87 | 0 | 86 | 86 |
| Abuakwa North (Kukurantumi) | 0 | 28 | 28 | 0 | 28 | 28 | 0 | 28 | 28 |
| **Total** | **36** | **317** | **353** | **37** | **316** | **353** | **35** | **316** | **351** |

compatibility of the innovation or evidence-based practice for a given practice setting, provider, or consumer; and/or perceived fit of the innovation to address a particular issue or problem, feasibility is the extent to which a new treatment, or an innovation, can be successfully used or carried out within a given agency or setting [20]. Community members from all six study communities perceived the CHAT intervention to be highly appropriate and feasible. This suggests that the CHAT intervention is a good fit with the value system of the community members and is regarded as practical and adequate for further implementation. Studies have shown that high appropriateness and feasibility are necessary requirements for successful intervention implementation and the achievement of intervention outcomes [19–21].

Furthermore, there was a statistically significant association between baseline and end-line assessment on all four effectiveness outcome measures. The study showed a high increment in awareness level, access, willingness to purchase and use of LLINs. The association between community involvement and malaria awareness is consistent with other malaria studies conducted in Rwanda and Malawi which found that community involvement in malaria control intervention increased awareness of malaria and the intervention being implemented [22,23].

**Table 4. Cross tabulations showing effectiveness outcomes from baseline to end-line.**

| | Time | | | |
|---|---|---|---|---|
| Responses | Baseline | Endline | Total | Chi-squared |
| **Awareness of LLINs** | | | | |
| No | 0 | 45 | 45 | |
| Yes | 11 | 744 | 755 | 19.45*** |
| Total | 11 | 789 | 800 | |
| **Access to LLINs** | | | | |
| No | 28 | 229 | 257 | |
| Yes | 44 | 499 | 543 | 124.02*** |
| Total | 72 | 728 | 800 | |
| **Willingness to Purchase LLINs** | | | | |
| No | 84 | 263 | 347 | |
| Yes | 126 | 327 | 453 | 47.55*** |
| Total | 210 | 590 | 800 | |
| **LLIN Use** | | | | |
| No | 106 | 313 | 419 | |
| Yes | 91 | 290 | 381 | 120.89*** |
| Total | 197 | 603 | 800 | |

***$p < 0.001$.

A high awareness level among community members could also help address community misconceptions about LLIN use as becoming knowledgeable about malaria prevention strategies will consequently increase LLIN use among community members [6]. Though there are few studies on willingness to pay for malaria control interventions in Sub-Saharan Africa [24,25], existing studies reveal that, when considered in the context of community engagement and the requirement for local support for financial sustainability of malaria control, the determination of consumer preferences and demand is significant in community-based health interventions [26,27].

Access to and use of LLINs has been vital in malaria prevention in Ghana. Previous studies have shown that there is a gap between LLIN access and LLIN use [6]. Evaluations have consistently discovered a large disparity between household ownership of LLIN and the use of LLIN, with LLIN use always significantly lower than ownership of at least one LLIN. This has been interpreted as evidence of the inability to achieve appropriate LLINs use or as a failure of behaviour change communication to sufficiently improve LLIN use [6,10,11]. By employing innovative social interventions that encourage behaviour change, and co-creating the CHAT interventions with community stakeholders, this study has recorded high access to LLIN and appropriate LLIN use in all six study communities. As evident in other community-based studies [22,23,25], engaging community members and key stakeholders at the grassroot level has shown to have contributed to a successful implementation of the CHAT intervention.

## Limitations

Although this study provided the implementation and effectiveness outcomes of CHAT intervention to improve LLIN distribution campaigns and continuous channels in six districts in Ghana with significant outcomes, the duration of the pilot implementation (i.e., 4 months) is relatively short to unravel all implementation and effectiveness outcomes. The distribution of the LLINS coincided with the activities of CHAT, as such, the observed increase in the implementation outcomes could be partly due to PMD. Also, the study was conducted in six communities within 2 regions in Ghana, so caution needs to be exercised when generalizing the findings to the whole of Ghana or other LMICs. Furthermore, due to the seasonality of malaria transmission in Ghana (i.e., during rainy seasons: April to June and September to November), the season within which the CHAT was implemented could have influenced the results. There is the need for long-term implementation of the CHAT to effectively observe the seasonal variations of implementation outcomes and effectiveness outcomes of CHAT to promote LLIN use for malaria control. This could be done by comparing implementation outcomes between communities with CHAT and those without. This would allow for difference in difference analysis.

## Conclusion

The CHAT intervention is effective in promoting LLIN use among community members. The use of SBCC, community mobilisation and stakeholder engagement has contributed to community involvement and buy-in of the CHAT intervention, thereby increasing its acceptability and feasibility in the study communities. The CHAT intervention is perceived as appropriate by community members and considered to be practical and adequate for further implementation. A higher level of awareness of LLINs recorded at the end-line will help contribute to the reduction of misconceptions regarding LLINs and malaria control and prevention, thereby increasing access to and use of LLINs. The high level of willingness to purchase LLINs is also important for local support of the financial sustainability of malaria control interventions. Finally, the CHAT intervention' effectiveness in increasing access to and use of LLINs could

contribute to malaria reduction and elimination in Ghana. The co-created CHAT in Ghana is an acceptable, appropriate, and feasible intervention for supporting the NMEP in the periodic Point Mass Distribution campaigns of LLINs, and for engaging in SBCC activities to support the continuous channels of distribution. CHAT can be used to transition the PMD and continuous distribution of LLINs into the Primary Health Care system in communities in Ghana.

## Supporting information

**S1 Checklist. Inclusivity-in-global-research-questionnaire.**
(DOCX)

## Acknowledgments

We would like to thank all stakeholders: members and staff from the Ghana Health Service (GHS) and the National Malaria Control Programme (NMCP), the Volta and Eastern Regional Directors of Health Services, the District Health Directors of Ho West; Ho; Agortime Ziope; Birim South; Achiase and Abuakwa North Districts, CHAT members, as well as community members who committed time to share experiences and provide data for this study. We are also grateful to the field staff for their meticulous work during data collection. Finally, we thank the HCE team (Allison Snyder, Patricia Richmond and Kristin Saarlas) for their constructive review and useful comments that strengthened the paper.

## Author Contributions

**Conceptualization:** Franklin N. Glozah, Philip Teg-Nefaah Tabong, Emmanuel Asampong, Adanna Nwameme, Philip Baba Adongo, Phyllis Dako-Gyeke.

**Data curation:** Emmanuel Asampong, Ruby Hornuvo.

**Formal analysis:** Franklin N. Glozah.

**Funding acquisition:** Franklin N. Glozah, Philip Teg-Nefaah Tabong, Emmanuel Asampong, Adanna Nwameme, Nana Yaw Peprah, Gloria M. Chandi, Philip Baba Adongo, Phyllis Dako-Gyeke.

**Investigation:** Franklin N. Glozah, Philip Teg-Nefaah Tabong, Eva Bazant, Emmanuel Asampong, Ruby Hornuvo, Adanna Nwameme, Nana Yaw Peprah, Gloria M. Chandi, Philip Baba Adongo, Phyllis Dako-Gyeke.

**Methodology:** Franklin N. Glozah, Philip Teg-Nefaah Tabong, Eva Bazant, Adanna Nwameme, Nana Yaw Peprah, Gloria M. Chandi, Philip Baba Adongo, Phyllis Dako-Gyeke.

**Project administration:** Eva Bazant, Ruby Hornuvo, Philip Baba Adongo, Phyllis Dako-Gyeke.

**Software:** Franklin N. Glozah.

**Supervision:** Franklin N. Glozah, Philip Teg-Nefaah Tabong, Emmanuel Asampong, Adanna Nwameme, Phyllis Dako-Gyeke.

**Validation:** Nana Yaw Peprah.

**Writing – original draft:** Franklin N. Glozah, Philip Teg-Nefaah Tabong.

**Writing – review & editing:** Eva Bazant, Emmanuel Asampong, Ruby Hornuvo, Adanna Nwameme, Gloria M. Chandi, Phyllis Dako-Gyeke.

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
