## [Decision Letter · Decision Letter 0]

28 Nov 2023

PGPH-D-23-01102

Implementation and effectiveness outcomes of Community Health Advocacy Teams to improve long-lasting insecticide net distribution and use in six districts in Ghana: a one-group pre-post-test study.

Dear Dr. Tabong,

Thank you for submitting your manuscript to PLOS Global Public Health. After careful consideration, we feel that it has merit but does not fully meet PLOS Global Public Health’s publication criteria as it currently stands. Therefore, we invite you to submit a revised version of the manuscript that addresses the points raised during the review process.

We look forward to receiving your revised manuscript.

Kind regards,

Ruth Ashton, Ph.D.

Academic Editor

Journal Requirements:

1. We noticed you have some minor occurrence of overlapping text with the following previous publication(s), which needs to be addressed:

- https://www.frontiersin.org/articles/10.3389/frhs.2022.1102328/full

In your revision ensure you cite all your sources (including your own works), and quote or rephrase any duplicated text outside the methods section. Further consideration is dependent on these concerns being addressed."""

2. Please include a complete copy of PLOS’ questionnaire on inclusivity in global research in your revised manuscript. Our policy for research in this area aims to improve transparency in the reporting of research performed outside of researchers’ own country or community. The policy applies to researchers who have travelled to a different country to conduct research, research with Indigenous populations or their lands, and research on cultural artefacts. The questionnaire can also be requested at the journal’s discretion for any other submissions, even if these conditions are not met.  Please find more information on the policy and a link to download a blank copy of the questionnaire here: https://journals.plos.org/globalpublichealth/s/best-practices-in-research-reporting. Please upload a completed version of your questionnaire as Supporting Information when you resubmit your manuscript.

Additional Editor Comments (if provided):

Reviewers' comments:

Reviewer's Responses to Questions

**Comments to the Author**

1. Does this manuscript meet PLOS Global Public Health’s publication criteria? Is the manuscript technically sound, and do the data support the conclusions? The manuscript must describe methodologically and ethically rigorous research with conclusions that are appropriately drawn based on the data presented.

Reviewer #1: Partly

Reviewer #2: Partly

2. Has the statistical analysis been performed appropriately and rigorously?

Reviewer #1: I don't know

Reviewer #2: No

3. Have the authors made all data underlying the findings in their manuscript fully available (please refer to the Data Availability Statement at the start of the manuscript PDF file)?

Reviewer #1: Yes

Reviewer #2: Yes

4. Is the manuscript presented in an intelligible fashion and written in standard English?

Reviewer #1: Yes

Reviewer #2: Yes

5. Review Comments to the Author

Reviewer #1: I read with interest this study entitled “Implementation and effectiveness outcomes of Community Health Advocacy Teams to improve long-lasting insecticide net distribution and use in six districts in Ghana: a one-group pre-post-test study”. Acknowledging the relevance of community-oriented approaches, the paper addresses a novel intervention aimed at promoting the utilization of Long-Lasting Insecticide Nets (LLIN) in selected districts in Ghana. This initiative involves the implementation of Community Health Advocacy Teams (CHAT), a strategy rooted in community engagement techniques. Beyond presenting this innovative approach within the context of LLIN use, the paper primarily focuses on sharing the results of a pre-post implementation study, that provides data on the positive outcomes of the CHAT intervention (specifically, its short-term effectiveness and some implementation-oriented outcomes such as acceptability, appropriateness and feasibility). The study's positive results suggest the potential for integrating the CHAT intervention into the Primary Healthcare system as a strategic program to enhance LLIN access and utilization.

While the specific methods employed by the authors fall outside my area of expertise, I have approached the paper with attention to the broader context, as I’m familiar with the research domain that the work explores. Hence, my feedback will be centred on study’s scientific relevance, conceptual coherence, and the clarity of its narrative.

Overall, the manuscript tackles an important aspect defining effective public health interventions, namely, community involvement in healthcare service delivery (specially regarding malaria prevention strategies). However, I noticed a few areas of the study assessing the CHAT short-term effectiveness and implementation outcomes that might require significant attention and revision before publication.

General comments:

In general, the paper addresses a topic of significant relevance, and the overall writing quality is commendable. However, upon careful review, it seems that certain sections and paragraphs might require revision to enhance readability. There are specific ideas that could benefit from improved linkage, ensuring greater clarity and fluency in the narrative. For instance, there are certain sentences in the 1st paragraph of the background section appears repetitive in their content.

The background should contextualise the study presented and should serve to prepare the reader understand the research gaps the paper aims to address. In this vein, the authors state that “more social, cultural, and behavioural research is needed to understand how local knowledge of transmission, diagnosis, treatment, and prevention influences the utilisation of ITN interventions” and “in order to create community-based malaria control programmes that are sustainable and encourage behaviour change as well as the adoption of new concepts and technologies, it is helpful to understand local knowledge”. While this argument is legitimate and supported by previous projects and accumulated evidence (and with which I agree), the study does not seem to cover these identified gaps, creating confusion for the reader. The argument is not revisited in the discussion, likely because the collected data, due to the study's objectives and design, do not allow for it. The authors might create certain expectations that are not fulfilled later on. I suggest rephrasing this important idea to integrate it better into the background section's narrative flow. For example, exploring how the CHAT intervention leverages local knowledge would make sense.

Another aspect that might require revision is the presentation of the CHAT intervention itself. Certain information appears to be missing (for instance, its origin: Who initiated this initiative? Is it part of a broader project? Is it affiliated with NMEP?). Reference number 13, cited in this subsection, pertains to a qualitative study that, presumably, is conducted by the same research team. Fortunately, the intervention is described more comprehensively in that paper. I strongly recommend that the authors revise this example. Additionally, providing more details on the co-creation process of the CHAT intervention would be highly appreciated.

One of the key concerns I have pertains to the clarity of the chosen conceptual framework guiding the assessment of the implementation outcomes. The authors state that they utilized the RE-AIM model, which apparently informed the selection and evaluation of effectiveness and implementation outcomes (i.e., acceptability, appropriateness, feasibility). However, given that the RE-AIM model comprises five dimensions pertinent to public health program planning and evaluation, none of which specifically address the selected implementation outcomes of this study (at least, not using the same terms), it would be helpful to provide details on how the model was employed and how it guided the assessment of short-term effectiveness and implementation outcomes. Specifically, clarifying the relationship between this conceptual model and the ultimately chosen outcomes is essential. A critical reflection, discussing the advantages and limitations of using the RE-AIM framework to assess implementation outcomes, would also be beneficial.

There are additional concerns regarding the methodology. Firstly, the authors state that this study follows a one-group pre-post design. I assume this means the same group of participants was involved in both phases of the study; however, the figures presented in Table 1 raise some doubts in this regard. I would appreciate clarification from the authors on this matter. Secondly, unlike the short-term effectiveness, there are no examples of the questions addressing acceptability, appropriateness, and feasibility. Given the complexity of defining and measuring these constructs, it would be valuable for the authors to provide some specific details. This will also allow to fully understand the results presented in the various tables.

Finally, in the discussion section, the authors address the crucial topic of trust-building, a key aspect often emphasized when evaluating the acceptability of a public health intervention at the community level. However, the connection between the collected data and the topic of trust is not sufficiently clear. I recommend that the authors elaborate more on this argument to enhance the understanding of how trust is implicated in their findings.

Specific comments:

Background:

• Line 71: Is the reference 1 cited correctly referred in the bibliography section? In case the authors wanted to cite the World Malaria Report 2022, the reference should be revised and corrected accordingly.

• Line 147-148: The acronym for the National Malaria Elimination Programme (NMEP) has already been used earlier in the text. There’s no need to use the long name again. I recommend the authors revise this throughout the text since there are other instances.

Methods

• Line 239: The 1st paragraph introducing the subsection on data collection appears to be confusing at this point in the text. I suggest moving it up to the introduction section, where the objectives of the study are being presented. This placement would provide better context and clarity for the readers.

Discussion:

• Lines 353-356: I’m not sure if the results provided helps support this statement. Can you elaborate more on this?

• Lines 394: The exact same sentence has been used earlier in the text. While I understand that the idea is being reintroduced, I recommend rephrasing it to create a more favorable impression on the reader.

I hope that above comments would help the authors to improve the manuscript. I am so looking forward to read the article once published.

Kind Regards.

Reviewer #2: The submitted manuscript describes implementation of a community health advocacy team (CHAT) and its impact on LLIN knowledge, attitudes, and practices in six districts of Ghana during a mass distribution campaign. Overall, the topic is intriguing, especially as it targets a real gap in the literature - namely how do we build consumer demand for LLINs, especially if and when support for public (i.e., free) distributions fades. Unfortunately, the study design suffers from significant methodological issues that call into question the validity of the results. Foremost among these is the concurrent PMD campaign. Because there is no control group (e.g., PMD only vs PMD+CHAT), one cannot disentangle any changes from the pre- to post-intervention period. For example, were participants more aware of LLINs and more likely to be using them because the PMD was ongoing and generally raised awareness or because of the specific effect of the CHAT intervention? For this reason, I do not think the results and interpretation - especially those relating to effectiveness - are valid. Additional comments below:

MAJOR

- The specifics of the CHAT intervention are not presented. Put simply, what did they actually do during the study period? How many meetings were there? How many household visits? There is no description of the activities conducted.

- There are standard WHO definitions of LLIN coverage measures including ownership, access, target coverage (1 LLIN per 2 HH members) and use, that are not referenced here and would have made for a more robust analysis. For example, we people not sleeping under LLINs because they did not have any LLINs, or they did not have enough LLINs for everyone in the household, or they had enough, but chose not to? Each is very different.

- The power calculation is not clearly described. What does a "50% anticipated use of mass LLIN distribution" refer to? A difference between pre- and post-surveys?

- Willingness to pay analysis is a discrete field of economics and requires much more than just a binary yes/no response. The results presented here don't truly capture that complexity and likely suffer from a social desirability bias.

MINOR

- Current accepted term is long-lasting insecticidal nets (LLINs), not insecticide treated nets

- Risk for severe disease (Line 77) is not just due to "lowered immunity", which is oversimplification

- Authors go back and forth between LLIN and ITN

- What does the proportion in each district mean? For example, Line 218. Is 90% and 75% the PfPR? If so that is an astounding level of transmission?

- Proportions should be presented in all tables

- Increasement (Line 332) should be increase

6. PLOS authors have the option to publish the peer review history of their article (what does this mean?). If published, this will include your full peer review and any attached files.

**Do you want your identity to be public for this peer review?** For information about this choice, including consent withdrawal, please see our Privacy Policy.

Reviewer #1: No

Reviewer #2: No

---

## [Editor Report · Decision Letter 1]

1 Feb 2024

PGPH-D-23-01102R1

Implementation and effectiveness outcomes of Community Health Advocacy Teams to improve long-lasting insecticide net distribution and use in six districts in Ghana: a one-group pre-post-test study.

Dear Dr. Tabong,

Thank you for submitting your manuscript to PLOS Global Public Health. After careful consideration, we feel that it has merit but does not fully meet PLOS Global Public Health’s publication criteria as it currently stands. Therefore, we invite you to submit a revised version of the manuscript that addresses the points raised during the review process.

We look forward to receiving your revised manuscript.

Kind regards,

Ruth Ashton, Ph.D.

Academic Editor

Journal Requirements:

2. We have noticed that you have uploaded Supporting Information files, but you have not included a list of legends. Please add a full list of legends for your Supporting Information files after the references list.

Additional Editor Comments (if provided):

Thank you for your revised version of the manuscript. I have a couple of final suggestions that could further strengthen your paper:

1. You mention several times the gap between LLIN access and LLIN use, suggesting that this is primarily a behavioral challenge. However, the indicator you have used to define access is a household with at least one LLIN, but where the household has more than 2 members this may not be "sufficient" nets for everyone to be able to sleep under a net. I suggest that this difference between having access to any net in a household and having sufficient nets is discussed at some point. You may also consider including the indicator of whether a household has at least one LLIN for every two members as an indicator of sufficient nets.

2. In the description of the study area you refer to the selected district has having highest prevalence, with a reference to the Ghana HMIS. However, routine HMIS data does not usually report prevalence since it is not a cross-sectional survey. Either this reference should be corrected to the MIS, or perhaps the actual indicator from HMIS used for selection of districts with incidence, or maybe test positivity rate among patients attending health facilities.

3. I would like to recommend including in the discussion a few sentences exploring whether the observed changes in your effectiveness outcomes are due entirely to the CHAT process, or partly a result of the point mass distribution occurring together with the CHAT. An alternative to this pre-post study design could have been to also conduct surveys in areas where PMD is taking place but without the CHAT component, allowing you to perform a difference in differences analysis and strengthen the plausibility that changes in your outcome indicators are attributable to the CHAT intervention.
---

## [Editor Report · Decision Letter 2]

4 Mar 2024

Implementation and effectiveness outcomes of Community Health Advocacy Teams to improve long-lasting insecticide net distribution and use in six districts in Ghana: a one-group pre-post-test study.

PGPH-D-23-01102R2

Dear Dr. Tabong,

We are pleased to inform you that your manuscript 'Implementation and effectiveness outcomes of Community Health Advocacy Teams to improve long-lasting insecticide net distribution and use in six districts in Ghana: a one-group pre-post-test study.' has been provisionally accepted for publication in PLOS Global Public Health.

Best regards,

Ruth Ashton, Ph.D.

Academic Editor